# The Analysis of Risk and Return Using Sharia Compliance Assets Pricing Model with Profit-Sharing Approach (Mudharabah) in Energy Sector Company in Indonesia

**Ahmad Faisol [1,2,\*], Sulaeman Rahman Nidar [1] and Aldrin Herwany [1]**

1 Department of Business and Management, Economic and Business Faculty, Universitas Padjadjaran, Bandung 40132, Indonesia
2 Department of Management, Economic and Business Faculty, Universitas Lampung, Bandar Lampung 35141, Indonesia
\* Correspondence: faisolkpm@gmail.com

**Abstract:** This study aimed to examine the relationship between risk and return using the Sharia Compliant Assets Pricing Model (SCAPM) with the profit-sharing approach (mudharabah) variable as a substitute for the risk-free rate (Rf) in energy sector companies in Indonesia as an empirical test object. The analytical tool used is univariate time series analysis using the ARX-GARCH model to determine validity of the model and forecast for the next 7 days. The findings showed a significant relationship between risk and return in a mining company in Indonesia. In addition, in terms of stock volatility, which is higher than market volatility, the shares of mining companies are shown to be in demand by investors compared to other average stocks in the Indonesian market. So, it can be concluded that the mudharabah variable can be used as a risk-free alternative rate (Rf).

**Keywords:** sharia compliance asset pricing; profit-sharing approach; ARX-GARCH model

## 1. Introduction

One of the analytical tools used to explain the relationship between risk and return of a security is the Capital Assets Pricing Model (CAPM) proposed by Sharpe (1964), Lintner (1969), and Mossin (1966), which explains the existence of a linear and significant relationship between returns security with a systematic risk level with a beta coefficient (β) and a return premium, namely the market return value after deducting the risk-free rate.

Several empirical studies support a positive and significant relationship between risk and return in the CAPM theoretical framework, including Pettengill et al. (1995) in Colombo, Lam (2001) in Hong Kong, Tang and Shum (2003), Sandoval and Saens (2004) in Latin America, and Xiao (2016) in the American stock market. In Indonesia, research that supports the CAPM theory or the significance of risk and return was put forward by Kisman and Restiyanita (2015) and Sembiring et al. (2016). However, research by Febrian and Herwany (2010), and Sutrisno and Nasri (2018) does not support the CAPM.

This difference in testing has also encouraged Islamic economic practitioners to examine the relevance of the CAPM model to Islamic investments. However, the use of a risk-free rate in CAPM model has caused debate among Islamic economics because it is considered usury (riba) and violates the principle of no profit without risk (al-ghunam bil ghurm). Therefore, they have tried to develop not only forms of sharia investment but also models of analysis and calculations that are subject to sharia principles, namely, eliminating the element of interest as a form of fixed income, which is considered usury or riba, as well as the imposition of zakat on income that is obtained.

Among the analyses that practitioners try to develop using these assumptions is an asset pricing model using the Sharia Compliant Assets Pricing Model (SCAPM) method. Previous researchers have tried several alternative methods by eliminating the element

of usury, including Tomkins and Karim (1987) who proposed removing all components of interest in all economic analysis and practice. This model was empirically tested by Hakim et al. (2016) in Malaysia, which compared the zero beta SCAPM and non-Rf SCAPM models with the classical CAPM model. This study found that both SCAPM models could explain sharia returns as conventional CAPM explained returns on all stocks.

Then, El-Ashker (1987) and Derbali et al. (2017) tried to develop the CAPM theory by replacing the risk-free rate with the zakat rate, which acts as the minimum rate of investment income or risk benchmark and purification of market returns. Furthermore, Shaikh (2010) proposes the use of the gross domestic product (GDP) rate as a substitute for the risk-free rate (Rf) because it is considered to represent the productivity of the community in an area. Islam recommends that everyone work productively and produce better values than previously. This productivity value can be used as an exposure risk of the expected profit resulting from work.

Next, Hanif (2011) proposes explicitly using the inflation rate to substitute for the risk-free rate (Rf) in the CAPM. This model is considered the most relevant model and approaches the classical CAPM model because the Rf component is formed by considering inflation in its determination. This model was then retested by Hanif and Dar (2013) by comparing the CAPM and SCAPM models, by using KSE-100 data for the period July 2001–June 2010. The findings are that there are no significant results for the high and low capitalization portfolio data in both models. Meanwhile, for portfolios with medium capitalization, the explanatory power of SCAPM is slightly better than CAPM.

Several other previous empirical studies that supported this research were also conducted by Subekti et al. (2020) who conducted research in Indonesia on the JII-70 index for the period January 2014 to December 2019. Using five SCAPM models, namely zero Rf, inflation, zakat, NGDP, and rate sukuk, this study found that SCAPM by using rate-sukuk was able to explain the level of the expected return is higher than other models. However, a significant test for comparing all sharia models has not been provided. Furthermore, Subekti and Rosadi (2022) also conducted an empirical test using the inflation SCAPM to test the performance of the sharia index stock portfolio in Indonesia during the COVID-19 period. Using the Black-Litterman (BL) method, this study compares the performance between BL-SCAPM and BL-CAPM with the result that during the COVID-19 period, the portfolio formed using BL-SCAPM is better than B-CAPM which is characterized by the impact of Sharpe ratio B-SCAPM is more significant, and portfolio losses are lower for the BL-SCAPM model. Furthermore, Rehan et al. (2021) conducted a comparison test between four SCAPM models, namely without risk free-rate, zakat, NGDP, and inflation, with conventional CAPM on the Pakistan Stock Exchange (PSX) for the period January 2001–December 2018 using the General Method of Moments. The test also considers the presence of an impact of size anomaly. The results of this study indicate that the SCAPM model with GDP and inflation has a larger significant value of explanatory power (R-square) and F-statistics compared to the conventional CAPM. Therefore, the two SCAPM models can be used to replace the CAPM in analyzing expected returns.

Although the model that was previously developed fulfilled several sharia principles, several things should be discussed, especially in its empirical application. The Askher and Derbali models that use zakat are considered to show a fundamental difference between zakat as a function of expenditure and Rf, which is a function of income. In addition, the provisions of the *haul* and *nisab* on zakat make this model difficult to apply. Likewise, the GDP model by Shaikh has several obstacles, namely, the difference in the principle of using the interest or risk-free rate as risk exposure, which is clearly different from GDP, where interest can be used as a monetary instrument to stabilize the economy; for example, when the government wants the economy to move more productively, then interest will be lowered, and vice versa, so that the movement between public interest and production (GDP) will be in the opposite direction. The inflation model by Hanif is considered a weakness because inflation is only one component in the formation of Rf, in addition to the cost of capital and the level of business risk. If only using inflation as a substitute for Rf,

the Expected Return (E(R)s) value of the Islamic CAPM will always be smaller than the classic CAPM.

Based on these criticisms, this study proposes the use of equivalent return of profit-sharing (mudharabah), which is a popular return on Islamic finance contracts, to substitute risk-free rate in the Sharia Compliance Assets Pricing Model or SCAPM. The SCAPM model using mudharabah has been proposed by Faisol et al. (2022) by comparing 6 SCAPM mudharabah models with conventional CAPM using the Mann–Whitney difference test. The findings obtained are that there is no significant difference in expected returns between the six variations of the SCAPM mudharabah model and the CAPM, so it can be concluded that mudharabah can be used in the asset pricing model.

Furthermore, this study will conduct an empirical test of the use of the SCAPM mudharabah model subject to zakah (SCAPM$_{RMDZ}$) to analyze the relationship between the risk and return relationship using ADRO from an energy company as a sample. The test was carried out using the ARX GARCH model, to form an optimal model for the relationship between risk and return. The data used are the return from the deposit of mudharabah contracts over a period of 12 months in Islamic banking in Indonesia. In addition, the calculation will also be subject to zakat as an instrument of purification assets, with the assumption that zakat is imposed on all income earned.

The scope of this research is limited to Islamic financial model, mostly in the asset pricing model.

## 2. Literature Review

### 2.1. Sharia Principle in Sharia Compliance Asset Pricing Model (SCAPM)

Sharia Compliance Asset Pricing Model (SCAPM) is an alternative model developed from the Capital Asset Pricing Model (CAPM) but assumes compliance with Islamic Sharia principles. These sharia principles include:

1. No interest charges The rule in Islamic economics prohibits the active practice of levy interest called usury (riba). The absence of this interest instruments creates its own problems in financial modeling based on sharia economics. On the other hand, interest can function as a return on investment. However, interest also acts as a risk measure for various investment alternatives which the projections are in the form of a risk-free rate (Rf). Based on the assumption that all investments are risky, the risk benchmark is the Islamic banking industry's return on the mudharabah contract. This legal contract fulfils the element of fairness in sharing profits and risks between related parties in an investment.

2. Imposition of zakat for income Zakat is the only financial instrument whose existence is ordered direct from God, as an obligation payment to pay net assets as well as a form of distributing wealth from people who have excess assets to those in need with certain criteria. Therefore, the SCAPM model will include zakat as an element of its calculation. The amount of zakat used is 2.5% of the profits obtained by a person.

### 2.2. Development of the SCAPM

The existence of the sharia principle has encouraged several researchers and practitioners of sharia economics to develop alternative models of CAPM that meet these rules. Several previous researchers have proposed instruments that can replace interest, including Tomkin and Karim, who proposed removing all components of fixed interest from the Islamic economy because the conventional economy has different traditions and laws from Islamic economics. Islamic economics prohibits fixed interest (riba), which adds to loans and certainty of profits on risky investments. Therefore, they propose that Islamic economic practices do not use fixed interest, and consequently, the CAPM model is considered invalid.

Ashker suggested replacing interest with zakat as the minimum return that investors must earn in investment to pay their obligations to their property. No minimum return investors will prefer to spend their money on consumption rather than investment. By

developing Ashker's notion of zakat, Derbali et al., with some modifications, reconstructed the SCAPM model by using a Sukuk or Islamic bonds as a risk-free replacement rate with the imposition of zakat on Sukuk and market returns as a fee charged for cleaning up assets treasure as required in the third rule.

Although the non-risk-free exchange rate model developed by Tomkin and Karim has complied with sharia principles by eliminating the risk-free exchange rate (Rf) function as a minimum return measure, there is no other alternative so it must be assumed that all asset investments are risky. Therefore, the CAPM model using beta is irrelevant.

So as with Ashker and Dirbali models, the difference between the functions Rf as income and zakat as a function of obligations or expenses also causes the zakat model to have weak assumptions.

Similarly, in the model developed by Dirbali, the use of the rate Sukuk as a substitute for Rf has a weakness because the nature of the Sukuk rate remains constant throughout the life of the Sukuk. So, its use cannot accommodate changes in economic and market conditions that occur during the economic life of the Sukuk. This situation causes the use of Sukuk can cause its own risk in its calculations.

Further, Shaikh argued that the use of interest with the principle of the time value of money is contrary to Islamic economics, which considers the increase in the value of money as a result of economic activity and not time. In addition, because every job, especially business, will always experience a business cycle, process risk-taking becomes commonplace. The use of interest that has provided certainty of return in investment is contrary to this principle.

Furthermore, Shaikh also stated that the use of interest in the economy would cause an imbalance in the market because interest can cause a shortage of capital. Therefore, it is essential to bring changes to practical finance according to Islamic economic principles. Shaikh proposed to replace the interest. Components with obligatory wealth, such as zakat, can be used as incentives for loans or other financing models so that creditors will still obtain a minimum return in wealth tax exemption. In addition, the elimination of interest also has the opportunity to increase trading activity because it can make money productive through trade and not loans so that investment increases and public debt decreases.

Another finding put forward by Shaikh is that there are indications that interest rates affect nominal GDP in the same direction. Therefore, GDP growth can be used as an assessment tool in Islamic finance models, including in the CAPM.

Shaikh's findings were later adopted by Mulyawan (2015), who used production growth (GDP) instead of Rf in its function as an exposed risk, with the argument that every investment is said to be productive when the income from an investment is above the growth of production/GDP in a region. This argument aligns with the belief that Islam recommends that income is the result of productivity and encourages everyone to work productively and provide better results than the previous time.

Although Shaikh stated that nominal GDP can be used instead of interest because it has the same direction as interest value. In theory, the focus of the work of the two is the opposite. With the increase in interest, the market will respond by reducing its products so that the value of GDP will fall, and vice versa. So, the use of GDP as a replacement no risk-free rate yet can be considered raw and still need tested level its validity with research other.

Next, Hanif tried to develop the SCAPM model by replacing the risk-free rate (Rf) with the inflation rate. The argument presented is because Rf contains two components: inflation and interest (actual Rf). Interest is forbidden in sharia principles. Therefore, inflation should be considered in predicting the level of profit in practical Islamic finance. Hanif calls his modified CAPM model Sharia Compliant Asset Pricing Model (SCAPM) with inflation.

Although considered the most relevant model, the model that uses inflation developed by Hanif has some weaknesses. The existing reality is that inflation is only one component in the formation of Rf besides the cost of capital and rate risk business. So, if the only use

of inflation is a replacement Rf, then the expected return value of SCAPM will always be smaller than the expected return of classic E(R) CAPM and not describe existing assets.

　　A brief explanation of the development of the previously proposed SCAPM model and some criticisms of it can be seen in Table 1.

**Table 1.** Development of the SCAPM.

| No | Author and Title | Result/Suggestion | Critics |
|---|---|---|---|
| 1. | Tomkins, C., & Karim, R. A The Shari'ah and Its Implications for Islamic Financial Analysis: An Opportunity to Study Interactions Among Society, Organizations, and Accounting. | 1. They suggest removing all use of interest in the sharia economy, both in practice and calculations, because of the differences in tradition and the basis for different laws between sharia and conventional economics. <br> 2. The CAPM model is considered invalid. Therefore, no alternative is needed. | Although moving from a different tradition and basic laws or philosophies, the CAPM model is considered as capable of helping market participants analyze and predict asset scores and therefore help market participants prudently manage their wealth and investments, following the maqashid sharia principles. Therefore, it is necessary to develop a suitable alternative CAPM model with sharia principles and not reject its use. |
| 2. | El-Ashker, AA-F The Islamic Business Enterprise. | 1. Propose zakat as a substitute for interest as a minimum return, with the hope that market participants prefer consumption to investment so that it can increase people's income/national income <br> 2. The proposed CAPM alternative models are: <br> $[E(_R) = 2.5\% + ((R_m - 2.5\%) + e]$ | Zakat and interest have different functions, where zakat is a function obligation or expenses, while interest or Rf is function income so the Ascher model has an error assumption because it becomes irrational if investors refuse investment that delivers profit below the zakat reckoning score (80 g of gold), however, could cover all cost or burden investment and still give profit. |
| 3. | Shaikh, SA Corporate finance in an interest-free economy: An alternate approach to the practice of Islamic Corporate Finance. | 1. Conducted research with 38 years of data from a group of large economies and found that nominal GDP is affected by changes in interest, and both move in the same direction. <br> 2. Proposes the use of GDP growth instead of Rf in the CAPM alternative model, with the following model form: <br> $[E(_R) = GDP + (R_m - GDP) + e]$ | In theory, the working direction of interest and GDP are opposite. An increase in interest rates usually the market will respond by lowering its productivity and switching to savings, so the value of GDP will fall, and vice versa. So, the use of GDP as a replacement has no risk-free rate yet and can be considered raw and still need to be tested for its level of accuracy. |
| 4. | Hanif, M. Risk and Return Under Sharia Framework: An Attempt to Develop Sharia Compliant Asset Pricing Model- SCAPM. | 1. Introduce the term Sharia Compliant assets Pricing Model (SCAPM) <br> 2. Propose charging the value of Rf with Inflation, arguing that the components forming interest are inflation and real Rf because real Rf contrary to sharia principles, inflation can be used to predict profit levels. The model developed by Hanif is as follows: <br> $[E(_R) = N + (R_m - N) + e]$ | Inflation is only one component in the formation of Rf besides the cost of capital and rate risk business. So, if the only use of inflation is a replacement Rf, then the expected return value of SCAPM will always be smaller than the expected return of classic E(R) CAPM and not describe existing assets. |

**Table 1.** *Cont.*

| No | Author and Title | Result/Suggestion | Critics |
|---|---|---|---|
| 5. | Derbali, A., El Khaldi, A., & Jouini, F. Shariah-compliant Capital Asset Pricing Model: New mathematical modelling. | 1. Propose rate Islamic bonds or Sukuk in the SCAPM model. With the following model form: $E_{(R)} = Rs + (Rm - \frac{Rs}{1-\partial Mt})$ where: $\partial_M$: The value of zakat as market purification 2. Develop a model for the imposition of zakat on sukuk returns and returns market as a fee charged for cleaning up assets, with the zakat model as follows: $\partial_M = 1 - \frac{\theta'\sigma M + (1–2.5\%)\ Rs}{(1–2.5\%)Rm}$ where: R: Return Sukuk; $\partial_M$: Market purification $\theta'$: Observed market price of risk on the capital market line; $\sigma_m$: market standard deviation | The model use of rate Sukuk as a substitute for Rf has a weakness because the nature of the rate of Sukuk is fixed throughout the life of the Sukuk, so its use cannot accommodate changes in economic and market conditions that occur during the economic life of the Sukuk. |

*2.3. State-of-the-Art*

Several criticisms of the previously developed SCAPM model have led this research to be interested in compiling the SCAPM using the average return equivalent to mudharabah as a risk-free replacement rate as a novelty. This mudharabah SCAPM modeling is built based on assumptions adopted from sharia economic principles.

The first assumption in model building is no risk-free rate in system Islamic economics which means all investments must be considered risky. Rf component in the CAPM must also be replaced with score payback other results that describe principle Islamic economics. This set average equivalent return mudharabah (RMD) as replacement Rf.

Based on the second assumption, the existence of the obligation of zakat on the value of the return, so the model of SCAPM mudharabah will become:

$$Ri = RMDZ_t + \beta\ (Rm - RMDZ_t),\tag{1}$$

Ri: Stock return on time i, RMDZ: return mudharabah with zakat on t period, β: beta or systemic risk, Rm − RMDZ: risk premium.

**3. Materials and Methods**

In this study, the data used were data on the stock returns of the Adaro energy company engaged in the field of energy from 2016 to 2020. In addition, daily return data from the JKSE market index and the equivalent return of profit-sharing (mudharabah) of Islamic banking in Indonesia are used for the same period. In conducting the analysis, the first step was to plot the time series data to see the behavior of the data. Then, in the second step, if the data were not stationary, data differencing was carried out, namely, changing the non-stationary data into stationary data. For stationary data, the Augmented Dickey-Fuller (ADF) test can also be used (Brockwell and Davis 2016; Tsay 2005). After the stationary assumptions were met, data modeling was carried out to obtain the best model. Before selecting the best model, we first found the optimal lag for the best model using the Akaike information criterion corrected (AICC) (Wei 2006). Based on the results of the AICC lag, the best model was determined based on the smallest value of the AICC. After the optimal lag was obtained for the best model, then model formation, parameter estimation, and hypothesis testing were carried out on the best model that was accepted. This study will

also examine the behavior of data return volatility. It will determine whether there is an autoregressive conditional heteroscedastic (ARCH) effect using the Lagrange Multiplier (LM) test (Tsay 2005). If there is an ARCH effect, then the modeling will be developed by including the residual modeling by applying the Generalized Autoregressive Conditional Heteroscedasticity (GARCH) so that the final model obtained is ARX(p,s)-GARCH(l,m) model. Based on this model, further analysis was developed to forecast the return data for Adaro Energy, Tbk stock.

### 3.1. Model Determination

The first step for analyzing the relationship between risk and return using the mudharabah SCAPM approach is to determine the suitable model for connecting both. A standard method for testing the relationship between risk and return is linear regression. However, this model must meet the assumption that homoscedasticity has no autocorrelation. If these two assumptions are violated, another model can be used, namely Autoregressive Conditional Heteroscedasticity (ARCH). For starters, Equation (1) can be converted into its linear regression model as follows:

$$Ri - RMDZ_t = \alpha + \beta(Rm - RMDZ_t), \tag{2}$$

$Ri - RMDZ$: excess stock return considering the mudharabah subject to zakat. Next will be called in notation Excess_RMDZ; $Rm - RMDZ$: risk premium considering the mudharabah subject to zakat. Next will be called in notation RISK_PRMDZ.

Furthermore, heteroscedasticity testing was carried out using scatterplots. The results of the calculations using can be seen in Figure 1.

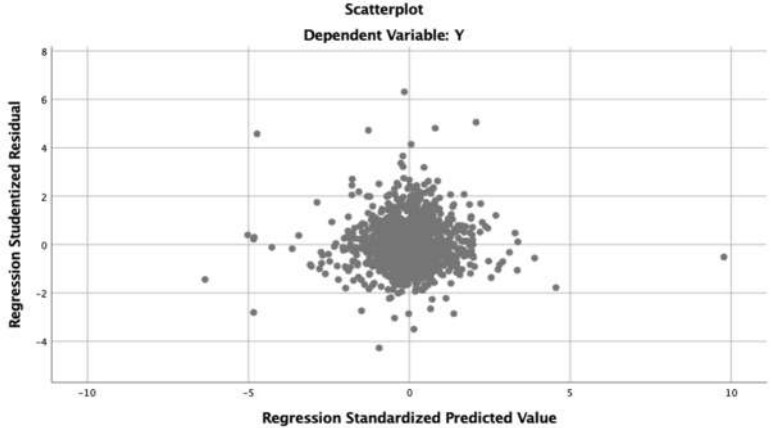

**Figure 1.** Heteroscedasticity test using a scatterplot.

Based on Figure 1, it can be seen that the scatterplot points are gathered around the zero line. Therefore, it can be concluded that there is heteroscedasticity, and a simple linear regression model cannot be used. Another model that can be used is the ARCH model.

### 3.2. Model ARX(p,s)-GARCH(l,m)

Based on Figure 1, the characteristics of stock returns are that their value is around zero, therefore it contains heteroscedasticity. Due to this, the model that will be used next is the ARCH model. Furthermore, it has been explained in Equation (2) that excess stock return considering mudharabah subject to zakat is denoted by Excess_RMDZ and risk premium considering mudharabah subject to zakat is denoted by Risk_PRMDZ.

RISK_PRMDZ$_1$, RISK_PRMDZ$_2$..., RISK_PRMDZ$_n$ are the time series data and Excess_RMDZ$_t$ following model ARX(p,s) with mean (μ), p is a lag from AR, and s is a lag from exogenous variable X. The mathematical model for the value of p = 1 and q = 1 could be written as follows:

$$\text{Excess\_RMDZ}_t = \mu + \Phi_1\text{Excess}_{t-1} + \Delta_1\text{RISK\_PRMDZ}_t + \Delta_2\text{RISK\_PRMDZ}_{t-1} + \varepsilon_t, \quad (3)$$

where $\varepsilon_t$ is white noise with mean 0 and variance $\sigma_t^2$ and $\Phi_1$, $\Delta_1$ also $\Delta_2$ is the parameter.

Model GARCH for $l$ = 1 and $m$ = 1 will be written as follows (GARCH (1,1)):

$$\sigma_t^2 = \beta o + \gamma_1\varepsilon_{t-1}^2 + \delta_1\sigma_{t-1}^2, \quad (4)$$

### 3.3. ARCH Effect Test

The autoregressive conditional heteroscedastic (ARCH) model was first introduced by Bollerslev (1986), where ARCH and GARCH predict variance is not constant depending on fluctuations in previous data. The GARCH model is a development of the ARCH model, which is widely used to estimate volatility (Engle 1982). The Lagrange Multiplier (LM) test is used to see if there is an ARCH effect on the return of PT. Adaro Energy tbk. The ARCH(p) model can be written as follows:

$$\sigma_t^2 = \vartheta + \sum_{i=1}^{p} \varsigma_i\varepsilon_{t=i}^2, \quad (5)$$

where $\varepsilon_t$ is the return on day $t$, $\sigma_t^2$ is the variance on day $t$, and $\vartheta$ and $\varsigma_i$ are positive constant values.

### 3.4. Forecasting

Forecasting data for the next 7-days was to be generated based on the best model obtained from the ARX(p,s)-GARCH(l,m) model.

## 4. Result and Discussion

However, it can be seen in Figures 2 and 3 that the excess return considering the mudharobah return subject to zakat or Excess_RMDZ (Figure 2) and the market risk premium considering the mudharobah subject to zakat or Risk_PRMDZ (Figure 3) shows that ADRO's stock return volatility fluctuated more than JKSE's market return. This fluctuation indicates that stock trading in the energy sector is quite high, causing high volatility. Next, if it is related to the concept of CAPM or SCAPM, the high volatility of the stock compared to the market indicates that the beta of the stock has a value of more than one, which means the stock has a high risk with the possibility of obtaining a high return.

Furthermore, Figure 2 shows that the Excess_RMDZ data fluctuate around zero indicating the data are stationary. So as Figure 3 shows, the data fluctuated slightly from 2016 to 2019, and in 2000 the risk premium was quite high. Risk premium data also fluctuates around zero, indicating that risk premium data are stationary. From Tables 2 and 3, the ADF test, the null hypothesis that the data are nonstationary was rejected, so based on the results of the ADF-test for data, Excess_RMDZ$_t$ and Excess_RM$_t$ are stationary. Therefore, the assumption stationary is fulfilled.

Tables 1 and 2 show the results of the ADF test (Tables 2 and 3). It can be seen that the Excess_RMDZ and Risk_PRMDZ data are stationary because the $p$-values (<0.0001) are both less than 0.05.

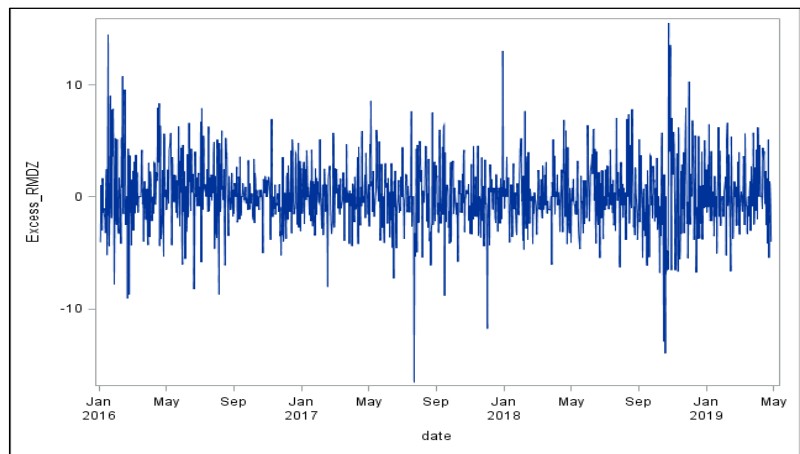

**Figure 2.** Excess return volatility data considering the return of mudharabah subject to zakat in 2016–2020.

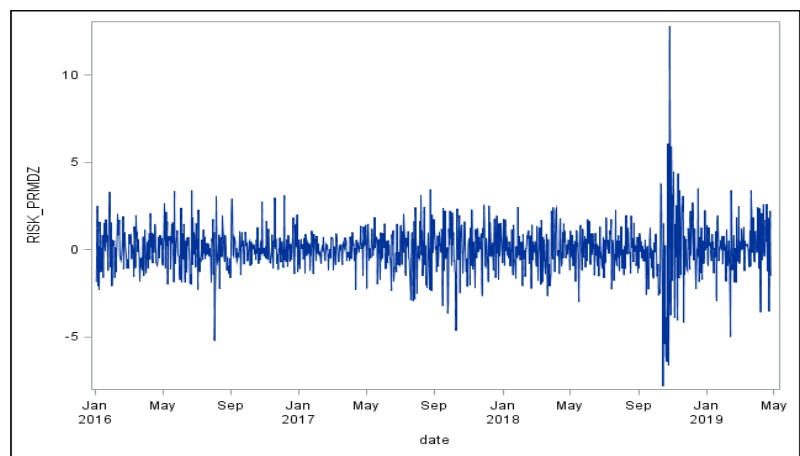

**Figure 3.** Risk premium market risk volatility data considering the return of mudharabah subject to zakat in 2016–2020.

**Table 2.** ADF test Excess_RMDZ.

| Augmented Dickey–Fuller Unit Root Tests | | | | | |
|---|---|---|---|---|---|
| **Type** | **Lags** | **Rho** | ***p*-Value** | **Tau** | ***p*-Value** |
| Zero Mean | 1 | −1255.71 | 0.0001 | −25.02 | <0.0001 |
| Single Mean | 1 | −1261.53 | 0.0001 | −25.07 | <0.0001 |
| Trend | 1 | −1268.00 | 0.0001 | −25.12 | <0.0001 |

**Table 3.** ADF test Risk_PRMDZ.

| **Type** | **Lags** | **Rho** | ***p*-Value** | **Tau** | ***p*-Value** |
|---|---|---|---|---|---|
| Zero Mean | 1 | −1425.80 | 0.0001 | −26.65 | <0.0001 |
| Single Mean | 1 | −1425.80 | 0.0001 | −26.64 | <0.0001 |
| Trend | 1 | −1426.42 | 0.0001 | −26.64 | <0.0001 |

Table 4 using the Ljung-Box test, shows that the Excess_RMDZ data has autocorrelation up to lag 18. This suggests that the Excess_RMDZ data modeling should involve autoregressive modeling.

**Table 4.** Autocorrelation check for white noise of Excess_RMDZ.

| To Lag | Chi-Square | DF | *p*-Value | Autocorrelations | | | | | |
|--------|-----------|----|-----------|------|------|------|------|------|------|
| 6 | 12.53 | 6 | 0.0512 | 0.016 | −0.029 | 0.093 | 0.014 | −0.016 | −0.009 |
| 12 | 26.59 | 12 | 0.0088 | 0.042 | 0.001 | −0.058 | −0.050 | −0.048 | −0.039 |
| 18 | 32.10 | 18 | 0.0214 | −0.013 | −0.047 | 0.012 | 0.000 | −0.002 | −0.045 |
| 24 | 33.17 | 24 | 0.1005 | −0.021 | −0.014 | 0.012 | −0.005 | −0.007 | 0.001 |

### *4.1. Autoregression Modeling*

The selection of model using Akaike Information Criterion Corrected (AICC) Criteria to check the optimum lag, Akaike's Information Criterion corrected (AICC) is carried out. Based on the results of the AICC analysis, the optimum lag for the AR(p) model is *p* = 1 (Table 5) because the value is smaller than the other values.

**Table 5.** AICC criteria.

| | Minimum Information Criterion Based on AICC | | | | | |
|-----|------|------|------|------|------|------|
| Lag | MA0 | MA1 | MA2 | MA3 | MA4 | MA5 |
| AR 0 | 1.9693522 | 1.971219 | 1.9706127 | 1.9713923 | 1.9728048 | 1.9711961 |
| AR 1 | 1.9692859 | 1.9717494 | 1.9720464 | 1.972855 | 1.9743718 | 1.9728601 |
| AR 2 | 1.9704537 | 1.9720928 | 1.9737019 | 1.974499 | 1.9759978 | 1.9745103 |
| AR 3 | 1.9715073 | 1.9726525 | 1.974297 | 1.9756928 | 1.9770181 | 1.9750703 |
| AR 4 | 1.9738123 | 1.9740115 | 1.9756416 | 1.9771297 | 1.9754315 | 1.9738063 |
| AR 5 | 1.9744295 | 1.9727598 | 1.9742084 | 1.9756467 | 1.9737452 | 1.9753051 |

### *4.2. ARCH Effect Testing*

A lot of data in the financial sector generally have a variance that is not constant over time. If the residual has a non-constant variance, then autoregressive modeling should also involve ARCH or GARCH modeling for the residuals (Tsay 2005; Wei 2006). Table 5 shows the results of the ARCH test with the null hypothesis that there are no ARCH effects. The test shows that the null hypothesis is rejected with F-test = 12.55 and *p*-value = 0.0004. Table 6 also shows the normality test using the Jarque_Bera (JB) test with the null hypothesis that the residual has a normal distribution. The results of the test show Chi-square = 563.14 with *p*-value < 0.0001, so the null hypothesis was rejected and the residuals are not normally distributed. However, Figure 4 shows that the distribution of error predictions and Q-Q plots shows that the deviation from the normal distribution is low, even though the statistical test shows that the null hypothesis was rejected.

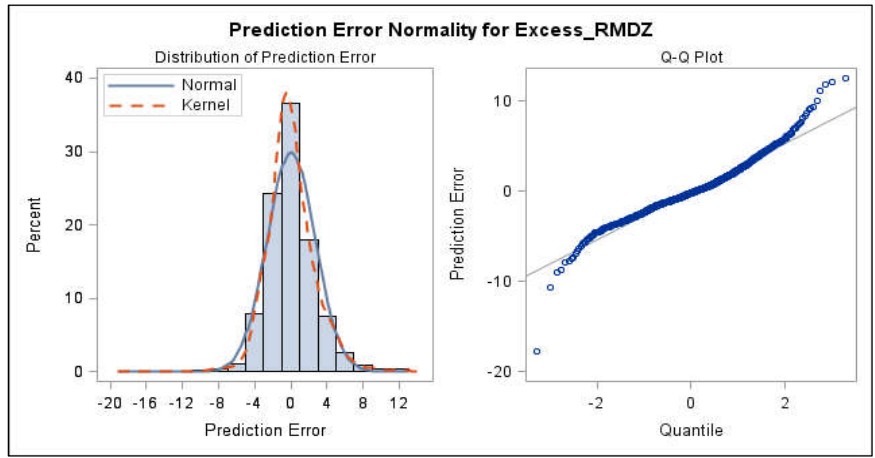

**Figure 4.** Normality error prediction for data Excess_RMDZ Adaro Energi Tbk.

**Table 6.** ARCH test for data Excess_RMDZ Adaro Energi Tbk.

| Variable | Durbin Watson | Normality | | ARCH | |
|---|---|---|---|---|---|
| | | Chi-Square | Pr > ChiSq | F Value | Pr > F |
| Excess_RMDZ | 1.99639 | 563.14 | <0.0001 | 12.55 | 0.0004 |

### 4.3. ARX(p,x)–GARCH(p,q) Model

Based on the results from the forecasting of the ARX (1.1) model in Table 7, the constant value of Excess_RMDZ is 0.08199, which means that if the other variables are zero (0), then the value of Excess_RMDZ is 0.08199. If the $RISK\_PRMDZ_t$ value increases by one unit, then Excess_RMDZ will increase by 1.14766. If the value of $RISK\_PRMDZ_{t-1}$ increases by one unit, then Excess_RMDZ will increase by 0.01754, and lastly, if the value of $Excess\_RMDZ_{t-1}$ increases by one unit, then Excess_RMDZ will increase by 0.01360.

**Table 7.** Parameter estimation model of ARX (1,1) data Excess_RMDZ Adaro Energi Tbk.

| Model Parameter Estimates | | | | | | |
|---|---|---|---|---|---|---|
| Equation | Parameter | Estimate | Standard Error | t Value | Pr > |t| | Variable |
| Excess_RMDZ | CONST1 | 0.08199 | 0.07088 | 1.16 | 0.2476 | 1 |
| | XL0_1_1 | 1.14766 | 0.06000 | 19.13 | 0.0001 | RISK_PRMDZ(t) |
| | XL1_1_1 | 0.01754 | 0.07151 | 0.25 | 0.8063 | RISK_PRMDZ(t−1) |
| | AR1_1_1 | 0.01360 | 0.03198 | 0.43 | 0.6708 | Excess_RMDZ(t−1) |

Based on the results of the analysis of the ARX (1,1)-GARCH (1,1), the forecasting model can be written using Equation (1): Average Model of ARX (1,1) based on Table 6:

$$Excess\_RMDZ_t = 0.08199 + 1.14766\ RISK\_PRMDZ_t + 0.01754\ RISK\_PRMDZ_{t-1} \\ + 0.01360\ Excess\_RMDZ_{t-1}, \tag{6}$$

Variance model of GARCH based on Table 8:

$$\sigma_t^2 = 0.81757 + 0.10462\ \varepsilon_{t-1}^2 + 0.78311\ \sigma_{t-1}^2, \tag{7}$$

From the parameter estimation of the GARCH model (Table 8), it can be seen that all the *p*-values are less than 0.05, so all the parameters are significant. Figure 5 shows conditional variance data Excess_RMDZ where conditional variance is highly relative in 2016, April 2018, November to December 2018, and relatively high on January to March 2020.

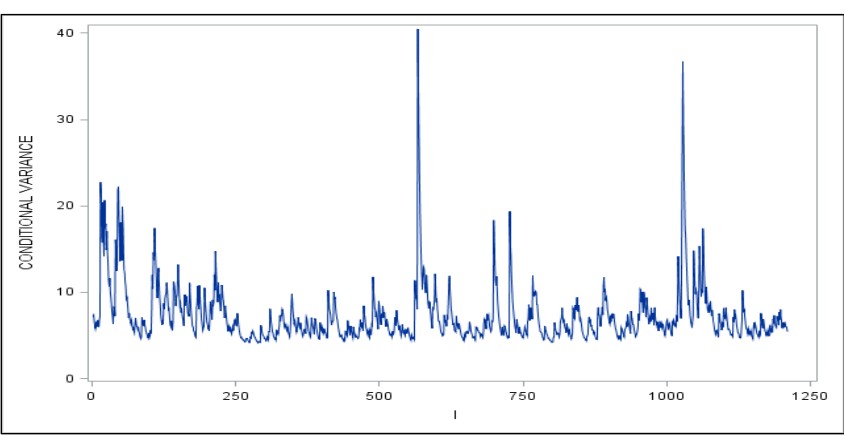

**Figure 5.** Conditional variance of data Excess_RMDZ.

**Table 8.** Parameter estimation model of GARCH (1,1) data Excess_RMDZ Adaro Energi Tbk.

| Parameter | Estimate | Standard Error | t Value | *p*-Value |
|---|---|---|---|---|
| GCHC1_1 | 0.81757 | 0.31031 | 2.63 | 0.0085 |
| ACH1_1_1 | 0.10462 | 0.03095 | 3.38 | 0.0007 |
| GCH1_1_1 | 0.78311 | 0.06603 | 11.86 | 0.0001 |

*4.4. Forecasting for Data Excess_RMDZ Adaro Energi Tbk*

Forecasting for the next 7-days of Excess_RMDZ data on Adaro Energi Tbk shares tends to be constant and there was no significant change (Figure 6). Table 9 shows that the first day up to the third day has increased. Meanwhile, for the 4th day to the 7th day, the forecasting value was constant. On the first day, the forecast value was 0.08105, on the second day, the forecast value was 0.11820, on the third day, the forecast value was 0.11988, while for the fourth day and so on, it was 0.1195.

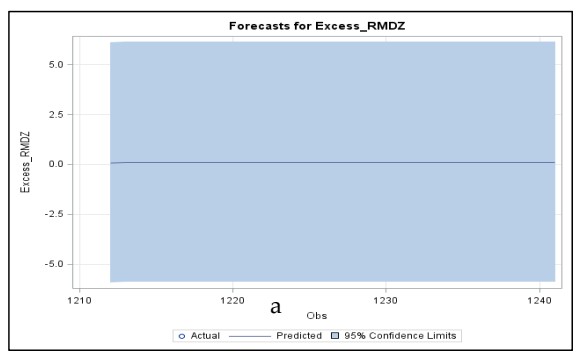 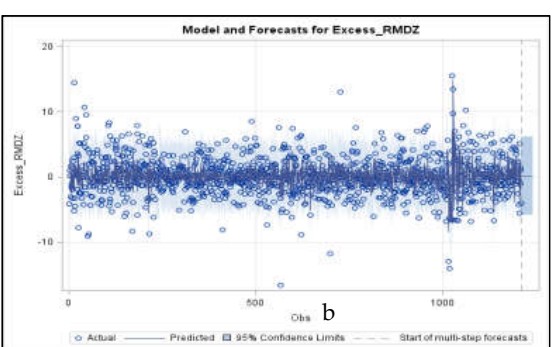

**Figure 6.** (**a**) Forecasting for data Excess_RMDZ and (**b**) model and forecasts for Excess_RMDZ Adaro Energy, Tbk.

**Table 9.** Data prediction of Excess_RMDZ Adaro Energi Tbk for 7 Days.

| Forecast Variable | Obs | Forecast | Standard Error | 95% Confidence Limits | |
|---|---|---|---|---|---|
| Excess_RMDZ | 1212 | 0.08105 | 3.06587 | −5.92794 | 6.09005 |
| | 1213 | 0.11820 | 3.06841 | −5.89577 | 6.13217 |
| | 1214 | 0.11988 | 3.06841 | −5.89410 | 6.13386 |
| | 1215 | 0.11995 | 3.06841 | −5.89403 | 6.13393 |
| | 1216 | 0.11995 | 3.06841 | −5.89403 | 6.13393 |
| | 1217 | 0.11995 | 3.06841 | −5.89403 | 6.13393 |
| | 1218 | 0.11995 | 3.06841 | −5.89403 | 6.13393 |

## 5. Conclusions

The results of testing the relationship between risk and return on ADRO energy sector stocks, as shown by a *p*-value of less than 0.05, indicate that by using the mudharabah SCAPM approach, the relationship between risk and return is significant. Thus, it can be concluded that mudharabah can replace the risk-free rate in the assets pricing model.

Meanwhile, the volatility of ADRO's stock is high compared to the volatility of its market return, which indicates that investors' interest in ADRO's stock is higher than the market average. Similarly, the results of the 7-day forecast post-calculation ending in December 2020 show that ADRO shares will experience high volatility in the first 3 days of forecasting; then, stock transactions will move to a constant condition. This means that not many stock transactions are carried out. It is possible that, in the first 3 days, investors hoped for a January effect from ADRO's stock returns, but in the following days, investors' responses returned to normal. The constant shape of the market is probably due to investors considering the effects of the COVID-19 pandemic on the market and refraining from making transactions.

## 6. Study Limitation

Although the SCAPM mudharabah model has been tested empirically in this study, there are some limitations in testing the model, including the use of the mudharabah variable, causing this model to only be used in countries that provide Islamic banking mudharabah equivalent data. Without it, mudharabah data needs require complicated calculations and take extra time.

**Author Contributions:** Conceptualization, A.F., S.R.N. and A.H.; methodology, A.F. and S.R.N.; software, A.F.; validation, S.R.N. and A.H.; formal analysis, A.F.; investigation, A.F.; resources, A.F.; data curation, A.F.; writing original draft preparation, A.F.; writing—review and editing, A.F.; visualization, A.F.; supervision, S.R.N. and A.H.; project administration, A.F.; funding acquisition, A.F. All authors have read and agreed to the published version of the manuscript.

**Funding:** The research received no external funding.

**Institutional Review Board Statement:** Not applicable.

**Informed Consent Statement:** Not applicable.

**Data Availability Statement:** Not applicable.

**Acknowledgments:** The authors would like to thank Universitas Lampung and Universitas Padjadjaran for the support with this postgraduate research study. The authors would like to thank to id.investing.com for providing data for this research. All authors acknowledge that this research has not been published in any journal or publication. In addition, this research is a self-funded research and has no conflict of interest with copyrights and other researchers. All authors respectfully participated in sharing ideas, methodology, and model enrichment of this research.

**Conflicts of Interest:** The author declare no conflict of interest.

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
