# Peer review of "The Analysis of Risk and Return Using Sharia Compliance Assets Pricing Model with Profit-Sharing Approach (Mudharabah) in Energy Sector Company in Indonesia"

_jrfm, doi:10.3390/jrfm15100421_

Round 1
Reviewer 1 Report (Previous Reviewer 2)
Thank you for offering me the the opportunity to review this manuscript. The topic is interesting, but limited to the Islamic area. Never the less it offers a picture of the model transposed to the local regulations. We are considering this perspective very useful for the theoretical enrichment.
Author Response
thank you for the review you gave. This model is actually in the area of ​​Islam because it was developed based on the principles of Islamic economics. however, the results of this model can be used empirically to assess the expected return of risky assets in general. Next I hope to get your approval so that I can publish this article on JFRM.
Reviewer 2 Report (Previous Reviewer 3)
The main aim of this study was to examine the relationship between risk and return using the Sharia Compliant Assets Pricing Model (SCAPM) with the profit-sharing approach (mudharabah) variable as a substitute for the risk-free rate (Rf) in energy sector companies in Indonesia as an empirical test object. Regarding the authors, I would like to congratulate and thank them for their effort and motivation involved in this research study. The presentation of the research is well documented, with a scientific basis. The methodology was chosen correctly. The conclusions support and result from the research and open new directions for future research. The submitted work is interesting, however, I do have a few comments that need to be answered, as well as a few thoughts that I think would enhance this article:
a) there are paragraphs in the introduction that do not refer to footnotes at any point. Please correct this.
b) although the bibliography has been strengthened (as I think, by highlighting), the reader may still feel uncomfortable about the small number of cited sources. If it is possible, add a few more sources so that the entire manuscript contains a minimum of 30 bibliographic references.
c) the bibliography must be aligned with the rules in the MDPI Instruction for Authors.
d) the results obtained in the original version should be included as supplementary material so that the reader can verify and comment on them.
e) the article needs to be supplemented with study limitations.
f) please add information on funding, author contributions and conflict of interest to the manuscript.
g) please separate the results from the discussion and treat them as two different chapters of the manuscript, reinforce the discussion further.
Supplementing the article with the above-mentioned scope will make in my opinion a real chance for publication in Journal of Risk and Financial Management. I strongly encourage the authors to continue this work during the revision and I keep my fingers crossed for the final success of the publication.
Author Response
thank you for the review you gave. Anyway, here's my answer to your comment:
a) I have checked and corrected several paragraphs so that they refer to existing references. And to ensure implementation, I use Mendeley as a software reference manager so that the same error does not happen again.
b) I have added the number of references that are considered necessary to support my research, so that it can be said to be feasible.
c) I have seen and studied the MDPI instruction. and based on my understanding, the bibliographic model used is the chicago model.
d) I don't understand the suggestion in point d, but I have resubmitted the results of the improvements I've made, so you can assess the progress of my work.
e) study limitations have been added as an additional chapter to the research paper.
f) information about funding and author contribution has been added to the acknowledgment
g)Referring to the JRFM template and other article examples, the results and discussion chapters are not separated. So that each section in the results chapter can be discussed.
Thank you for your comment.

Reviewer 3 Report (New Reviewer)
The reviewer’s comments are as the followings.
1. The authors describe a modeling of risk analysis for Indonesia’s energy sector company. The background and literature survey are well explained but they should be improved by updating the review up to present (2022).
2. The objectives of this research work are clear and consistent to the methodology. Scopes of this research work should be also mentioned in the last paragraph after the objectives.
3. In Abstract, the model test conditions for the time series analysis using the ARX-GARCH model should be briefly described.
Author Response
thank you for the review you gave. here are some of my answers to your comments :
1. I have updated the background and literature used, one of which uses the 2022 reference.
2. I have also added the scope of the research in the background.
3. In the abstract, I have tried to provide additional explanation about ARCH-GARCH.
Thank you.

Reviewer 4 Report (New Reviewer)
Thank you for giving me the opportunity to read and reflect on your paper, I think this is well structured and well written. It is a good paper.However in my opinion, some important details should be emphasized in order to improve the quality of the research before publishing:
The literature review is not the most appropriate. There are more recent works that should be cited. The most current one that appears in the text, is from 2020. It is mandatory to improve this aspect.
The conclusion section should connect the research results with relevant literature citations for validity and reliability
Author Response
thanks for your suggestion, and I've added some additional literature with the latest year as you suggested.

This manuscript is a resubmission of an earlier submission. The following is a list of the peer review reports and author responses from that submission.
Round 1
Reviewer 1 Report
The text on lines 86 through lines 100 should be dropped.
Equation 107 is missing a "+" sign
The notations RISK_PRMDZ_i and Excess_RMDZ_t (line 103) are used before they are defined (lines 137 and 138).
In the equation on line 107, Excess_RMSZ_t on the left hand side of the equality is written with RMDZ not subscripted, while on the right hand sideRMDZ is subscripted. The absence of subscripts also occurs in line 103. All variables should be written consistently.
Line 109 does not clearly identify \epsilon_t as the white noise variable. As written, it currently implies Excess_RMDZ_t is white noise.
Replace the word "variant" by "variance" everywhere in the manuscript.
Lines 115 and 118 refer to numbered references [9] and [8]. The references are not numbered.
The author's claim that Figure 1 shows "between September 2016 and January 2019, high volatility was experienced compared to the other days." (line 139) is not substantiated by the data.
The majority of the analysis is comprised of standard fitting methods to an ARX-GARCH model.
I could not find an indication of how many trajectories were utilized in the author's 30 day forecast. Forecasting using a GARCH model must be considered suspect. I refer the author's to the paper "GARCH(1,1) process can have arbitrarily heavy power tails", Lithuanaian Math. Journal. 4792):164-175. DOI:10.1007/s10986-007-0012-z by A. Klivecka and D.. Surgailis. Generally such forecasts will generate confidence limits that are so huge as to mask any signal. This appears to be the case in this study.
Author Response
please see the attachment.
I have revised the article as you suggested. I hope it matches or at least is close to what you expect. Thank you

Reviewer 2 Report
Thank you for offering the opportunity to review this manuscript. Thera are few aspects that has to be improved:
- a subchapter "Literature review" has to be developed after introduction and present the similar studies, the methods used, and the results obtained;
- the references has to be extended to cover at least the relevant studies about the relation between risk - revenue or risk - ...., revenue - .....
- than, the gap addressed by the paper, research question and hypothesis has to be clearly mention
- the conclusion has to be developed to provide information haw the proposed model contributes to the economic science and practice;
- the abstract has to be restructured in accordance with the correction of the paper.
Author Response
please see the attachment.
I have tried to revise the article as you suggested. I hope it matches or at least is close to what you expect. Thank you

Reviewer 3 Report
The main aim of this study was to examine the relationship between risk and return using the Sharia Compliant Assets Pricing Model (SCAPM) with profit-sharing approach (mudharabah) variable as a sub-stitute for the risk-free rate (Rf) in energy sector companies in Indonesia. Regarding the authors, I would like to congratulate and thank them for their effort and motivation involved in this research study. Although, this field of research is interesting, as explained in the work under study, the paper requires clarification of certain issues.
The introduction is a bit chaotic. What does the first short sentence ‘Pricing Model (SCAPM) method.’ symbolize? In such a topic, I would expect a consistent step-by-step explanation of the concepts contained in the title, such as risk and return analysis or sharia compliance asset pricing, and then the development of a thematic thought that will smoothly pass to the goal of the work, and then encourage you to read the further methodological and discussion.
The ‘Materials and Methods’ section needs to be completely reworked. Lines 85-100 should not appear in the finished manuscript for review, as they indicate the sloppy attitude of the authors to the research and the journal. The methodology should be described with sufficient detail to allow others to replicate and build on published results. Divide the methodology section preferably into subsections of Design, Procedure, Measures and Data Analysis. The methods and protocols should be described and appropriately cited. In addition, the MDPI’s style for citations and references lists are widely based on the style used by the American Chemical Society. Please refer to the ACS Style Guide and customize your entire manuscript.
In the ‘Results and Discussion’ section, there is basically no discussion, the main focus is on presenting the results. This cannot be the case, and your results should be related to other existing studies. It would also be useful to know if similar studies were carried out outside Indonesia and what their results were.
While I really applaud the interesting topic, this manuscript requires a major revision from the authors in order to make a meaningful contribution to the literature. I am aware that the text will have to be practically fully revised in order to be published in the Journal of Risk and Financial Management. However, I strongly encourage the authors to continue this work.
Author Response
Please see the attachment.
I have tried to revise the article as you suggested. I hope it matches or at least is close to what you expect. Thank you

Round 2
Reviewer 1 Report
See attached

Reviewer 2 Report
The authors improved the manuscript.
Reviewer 3 Report
Thank you for the revised manuscript. While I appreciate the authors' input and commitment, in my opinion there are still many issues that need improvement, as I mentioned in my previous review. The current version of the article still does not include discussion, which is a must for such studies. The section covering methodology should preferably be divided into subsections Design, Procedure, Measures and Data Analysis. Methods and protocols should be described and properly cited, confirming their use in previous studies. In addition, the MDPI style for citations is still not applied, please refer to the MDPI Instructions for Authors and adapt your entire manuscript. Although I appreciate some of the changes made by the authors, I think that the article still needs a lot of work before its final publication in the Journal of Risk and Financial Management.